# Therapeutic Applications of Aptamers

**DOI:** 10.3390/ijms25126742

**Published:** 2024-06-19

**Authors:** George Santarpia, Eric Carnes

**Affiliations:** College of Public Health, University of Nebraska Medical Center, Omaha, NE 68198, USA

**Keywords:** aptamers, SELEX, nucleic acid therapeutics

## Abstract

Affinity reagents, or target-binding molecules, are quite versatile and are major workhorses in molecular biology and medicine. Antibodies are the most famous and frequently used type and they have been used for a wide range of applications, including laboratory techniques, diagnostics, and therapeutics. However, antibodies are not the only available affinity reagents and they do have significant drawbacks, including laborious and costly production. Aptamers are one potential alternative that have a variety of unique advantages. They are single stranded DNA or RNA molecules that can be selected for binding to many targets including proteins, carbohydrates, and small molecules—for which antibodies typically have low affinity. There are also a variety of cost-effective methods for producing and modifying nucleic acids in vitro without cells, whereas antibodies typically require cells or even whole animals. While there are also significant drawbacks to using aptamers in therapeutic applications, including low in vivo stability, aptamers have had success in clinical trials for treating a variety of diseases and two aptamer-based drugs have gained FDA approval. Aptamer development is still ongoing, which could lead to additional applications of aptamer therapeutics, including antitoxins, and combinatorial approaches with nanoparticles and other nucleic acid therapeutics that could improve efficacy.

## 1. Introduction

While the primary structure of nucleic acids is important for their role in maintaining and relaying genetic information, the secondary and even tertiary structures of single stranded nucleic acids can also have important functions [1]. This structure is primarily mediated by base pairing between different regions of the molecule, which can result in a wide variety of structures, the classification and study of which is an active area of research [1,2]. These structures can allow some nucleic acids, called aptamers, to bind other molecules. As nucleic acids contain relatively hydrophobic and planar bases, hydrogen bond donors and acceptors in the bases and sugar, and a charged phosphate backbone, aptamer–ligand interactions can be stabilized through stacking interactions, hydrogen bonding, and electrostatic interactions between charged atoms [3,4,5,6]. These interactions are also facilitated by the flexibility of nucleic acids, as six of the bonds in the phosphate backbone, as well as the glycosidic bond to the sugar, can rotate [7]. Additionally, steric hindrance between the aptamers and other molecules can be an important contributor to specificity [8]. Interestingly, though, the specificity of aptamers does not correlate well with their affinity, meaning higher affinity aptamers may not necessarily be more specific for the target [9]. This unusual property indicates that different aptamers have unique interactions with their ligand, resulting in independent affinity and specificity patterns [9]. Ligands that are smaller than the aptamers are typically enveloped upon binding [3,4,5,6,8] and aptamers that are smaller than their ligands typically either attach to the surface or are incorporated into the structure of the ligand [7]. Aptamer–ligand binding can also lead to conformational changes in the aptamer, target, or both, and this induced fit can contribute to binding strength [8,10,11]. A few example structures of aptamers binding their ligands are shown in Figure 1. Far from being a simple curiosity, aptamers serve important functions in nature, with the most well-known examples being riboswitches. Riboswitches are typically found at the 5′ end of bacterial mRNAs and are composed of an aptamer region and expression platform upstream of the coding region [12]. Upon ligand binding, the secondary structure of both the aptamer and the expression platform changes, which can alter transcription termination or the accessibility of the ribosome binding site [12].

Aside from their roles in nature, aptamers are important tools in biochemistry and biomedical sciences. Like antibodies and other affinity reagents, aptamers have a variety of applications, including in imaging and as biosensors [13]. They also show significant promise as therapeutic agents and diagnostics for a wide range of diseases [14,15]. There is an expansive body of literature in this area, so this review aims to provide examples of key methods involved in selecting and producing target-specific aptamers, emphasize important advantages and disadvantages when compared to antibodies, discuss the aptamers that have been approved by the FDA or are in clinical trials, and highlight several promising research directions for the development of new aptamer-based therapeutics. 

## 2. Aptamer Selection and Production

### 2.1. Aptamer Selection

The most common method of obtaining target-specific aptamers is the systematic evolution of ligands via exponential enrichment (SELEX; Figure 2). Randomized pools of oligos with primer binding sites are commercially available and necessary for this process. The first step is to separate oligos that bind the target of interest from ones that do not. Then, the target-binding aptamers are amplified using polymerase chain reaction (PCR)-based methods and the process is repeated. Additionally, oligos that bind to unwanted targets can be removed through a similar process called negative selection or counterselection. Once this process has been repeated enough times, the oligos can then be sequenced and the individual aptamer sequences can be tested against each other for affinity and specificity. Examples of different methods for this process will be highlighted below, though a full list of the possible variations is beyond the scope of this review.

There are a wide variety of methods to separate target-binding from non-binding oligos. Traditional methods involve affixing the target to solid surfaces, such as nitrocellulose filters or beads. Nitrocellulose filters are useful when the target is a protein or other large molecule, as those targets and any aptamers bound to them are unable to pass through the filter while the unbound aptamers can, but they are more difficult to use for smaller targets due to pore size limitations [16]. For the beads, oligos that bind the target can be separated by a variety of methods, including affinity chromatography with bead-packed columns [17] and magnetism [18], depending on the type of beads used. A disadvantage of these methods is that they tend to require large amounts of the target, though microfluidic columns can be used with the target-affixed beads to solve this issue [19]. However, immobilization-based techniques have the disadvantage of interfering with the interactions between the oligos and target. The immobilization process can change the conformation of the target [20], the surface can cause steric hindrance preventing the binding of the oligos and provide non-specific binding sites [21], which can result in the identification of fewer target-binding oligos and more oligos that cannot bind to the native target. To address this, methods have been developed to use solubilized target molecules, some of which involve affixing the oligos instead. One such method is Capture-SELEX, in which the ability of aptamers to change conformation upon target binding can be used to remove target-binding aptamers from the surface they are affixed to, though this method is typically limited to small molecule targets [22]. Another approach is to have both the oligos and the target in solution. The most common method for this is capillary electrophoresis or CE-SELEX, which takes advantage of the change in electrophoretic mobility upon target binding [21]. This approach can be used to identify target-binding aptamers after a single round of selection, in a process known as non-SELEX [23], but it has the disadvantage of requiring expensive equipment that limits the amount of oligos that can be screened, due to limited capillary loads [21]. Finally, FACS sorting and other methods can be used to conduct the selection process using cells, tissues, or even in live animals, which has the advantage of ensuring the target is in a physiologically relevant state and negative selection can be used to ensure that non-specific aptamers are not selected [24].

Once target-binding aptamers are separated from the non-binding oligos, they typically must undergo rounds of amplification in order to conduct subsequent rounds of selection. This process is often facilitated by identical primer binding sites on all of the oligos in the pool, which allows for PCR-based amplification [21]. For DNA aptamers, asymmetric PCR is a common approach for amplification, though methods for separating the strands in the dsDNA copies also exist [21]. Asymmetric PCR involves increasing the ratio of one primer over the other, which results in the normal exponential amplification of both strands, followed by linear amplification of the strand associated with the more abundant primer, thereby producing single-stranded products [25]. While this method does require optimization, it can efficiently produce ssDNA in a single reaction and optimization pipelines have been developed for aptamer selection [25]. For RNA aptamers, the process is more complex. First, the RNA must be converted to DNA and amplified using RT-PCR, adding a promoter sequence in the process [21]. Then, the dsDNA copies must be transcribed into RNA using in vitro transcription, resulting in ssRNA copies of the original aptamers [21]. Due to the time-consuming nature of this process, a method known as RAPID-SELEX (RNA Aptamer Isolation via Dual-cycles SELEX) has been developed to streamline the process by reducing the number of amplification steps necessary for RNA aptamer selection [26]. Once the aptamers have been appropriately selected and amplified, they can then be sequenced to identify the collection of sequences capable of binding the target.

### 2.2. Binding Analysis and Optimization

Once the target-binding aptamers are sequenced, the individual aptamers can be produced, and their affinity and specificity can be measured. A traditional method for this is an electrophoretic mobility shift assay or EMSA, which relies on the difference in electrophoretic mobility between the aptamer-target complex and the aptamers or target alone [21]. Dot-plots are an alternative assay where the interaction between the aptamer and target results in a colorimetric or fluorescent signal [21]. A more modern approach is the fluorescence/Forster resonance energy transfer (FRET) assay, which involves linking both the target and aptamer with fluorophores [27]. The donor fluorophore fluoresces on its own but transfers its energy to the acceptor when they are in close enough proximity, causing the acceptor to fluoresce [27]. The emissions from the donor and acceptor fluorophores can then be used to calculate the binding constants for the interaction between the aptamer and the target [27]. Once the binding behavior of each aptamer is measured and the best performing aptamers are chosen, the aptamers can then be optimized.

A common first step in aptamer optimization is optimizing the length by removing unnecessary sequences, which allows for a more efficient and cost-effective production of the aptamer [21]. The oligo libraries used during the SELEX process include primer binding sites on both ends that frequently do not participate in the binding interaction, so they can be removed without affecting binding affinity in most cases and, in some cases, removing them can even improve affinity [28]. Other sequences in the aptamer can also be removed and the resulting oligos can then be tested to determine if their removal affected the binding affinity. This process can be aided using software that predicts nucleic acid secondary structures and aligns consensus sequences. Once the length is optimized, chemical modifications to the bases, backbone, and ends of the nucleic acid can be tested. To increase the structural diversity and maximize the potential binding interactions, chemically modified nucleic acids can be tested, either with modified libraries during the selection or modification of selected aptamers after SELEX. There are also a number of ways in which the stability of aptamers can be increased. Some modifications—such as the replacement of the phosphodiester backbone, additions to the 2′ or 4′ carbon of the ribose, and the capping of the 3′ or 5′ end—decrease recognition by nucleases [29,30]. Additionally, L-nucleic acids can be used instead of the natural D-nucleic acids to produce aptamers—known as Spiegelmers—that are not recognized by nucleases due to mismatches in orientation [31]. The ribose in the nucleic acid backbone can also be replaced with a different sugar or cross-linked, which leads to increased nuclease resistance and half-lives, as well as enabling the formation of unique secondary structures [32,33]. Additionally, large compounds like polyethylene glycol can be added to the 5′ ends of the aptamers to prevent the loss of aptamers through the kidneys when administered to live animals [34]. While these modifications can be incredibly useful in designing aptamers for therapeutic use, they can also lead to issues with PCR amplification, as natural polymerases often cannot recognize the modified bases used or generate the unnatural backbones. However, mutant polymerases capable of recognizing unnatural bases have been developed to circumvent these issues [35] and the modifications can be applied after PCR amplification.

### 2.3. Methods of Production

As with the amplification of oligos during SELEX, PCR-based methods can be used to produce aptamers, including asymmetric PCR to produce DNA aptamers and in vitro transcription to produce RNA aptamers. However, there are a few limitations to these methods. For one, reaction conditions need to be optimized to produce each aptamer, especially with asymmetric PCR [25]. Additionally, the oligo primers still need to be synthesized for DNA aptamers, so the cost effectiveness of this method would depend on the length of the aptamers relative to the primers. This is exacerbated by the fact that asymmetric PCR produces dsDNA copies that cannot be used [25]. Furthermore, as mentioned earlier, chemical modifications to the aptamer may cause complications in production using PCR.

Another option for production is solid-phase synthesis. Nucleoside phosphoramidites are commonly used for this, as they are modified nucleotides with a protecting group that prevents unwanted reactions that would alter the sequence of the oligo [36]. As indicated by the name, this method also relies on a solid support to which the growing nucleic acid chain is attached. The process involves a cycle of deprotection of the nucleoside attached to the support, the addition of a new nucleoside phosphoramidite corresponding to the next nucleotide in the sequence, the coupling of the two nucleosides, oxidation or sulfurization to produce more stable intermediates, and finally the capping of any unreacted nucleosides with an acetyl group to prevent them from participating in subsequent synthesis steps [36]. The process can then be repeated with the next nucleoside until the full-length oligo is produced. The resulting polymers are then cleaved from the support and their unnatural chemical modifications are removed [36]. With the introduction of microchip-based methods and other improvements, hundreds of different oligos can be produced in parallel, with costs between USD 0.00001 and 0.001 per nucleotide synthesized, though the exact costs and scales vary by vendor [37].

## 3. Comparison to Antibodies

Antibodies are standard affinity reagents that are used in a variety of applications, from common biochemical techniques like Western blotting [38], to disease diagnostics [39], and treatments [40]. However, while they are incredibly useful, their production can be costly and cumbersome. The most common approach for generating antigen-specific antibodies is through immunizing live animals, often rabbits or mice, with that antigen and collecting their serum after the animals mount an effective humoral response [40]. For therapeutic use, humanized mice with human antibody sequences can be used to generate humanized antibodies that are not as immunogenic [41]. Antigen-specific antibodies can then be purified, with methods like affinity chromatography, to make polyclonal antibodies [42]. While this procedure is relatively straightforward, it requires the use of animals for every batch. For human diseases, convalescent serum, plasma, or whole blood may also be collected from survivors of that disease and used to treat subsequent patients. However, this is usually only used in emergencies, such as during the COVID-19 pandemic, due to batch-to-batch variability and the need for blood group matching [43]. Alternatively, antigen-specific B cell clones can be purified and fused to an immortalized cell, generating a hybridoma that can have a long-term expression of the monoclonal antibody that the B cell produced [44]. Additionally, the antibody genes can be sequenced, allowing non-human animal constant regions and framework regions to be replaced by the human counterparts, in a process known as complementary-determining region (CDR) grafting [45]. However, obtaining the B cells still typically requires animal immunization and the work to generate and maintain the hybridomas and purify the antibodies can still be expensive, especially relative to aptamers, which can be made synthetically. Additionally, unlike with aptamers, this method is difficult to use with targets that are not immunogenic, as they typically need to be conjugated to carriers that can interfere with antibody binding [46], or are toxic to the animals, as they would need to be administered in low doses.

An alternative approach that can circumvent the need for live animals in production is to use phage display [47]. In this, a library of antibody sequences can be fused to the capsid gene of a bacteriophage and inserted into a phagemid that allows the production of the phage displaying the antibody sequences [47]. This can be carried out with Fab fragments [48] or single-chain variable fragments (scFvs) [49]. The target can then be used to separate out the phage with sequences that bind it, and those phages can then be propagated by infecting their bacterial host. This process is repeated until adequate binding is obtained and then the phage genomes can be sequenced to determine the sequences that bind to the target. This circumvents many of the issues with non-immunogenic and toxic targets but issues with cloning and expression can remove high-affinity antibody sequences from the pool [50]. Additionally, the large-scale production of these antibodies or antibody fragments still involves purification from cell culture.

In addition to the cost of production and the ability to be made entirely in vivo, aptamers have several other advantages over antibody technologies. First, they are stable at room temperature for years and can renature after heat denaturation [51], unlike antibodies that typically require −80 °C for long-term storage and can permanently denature and aggregate at higher temperatures [52]. This is especially advantageous for applications like biodefense that require stockpiling, or access to developing countries without adequate cold chains. Aptamers are also much smaller than antibodies and are capable of crossing physiological barriers such as the blood–brain barrier [53,54] more efficiently, improving the delivery of therapeutic treatment. Additionally, aptamers tend to have a better affinity for small molecules than antibodies. For example, multiple aptamers with nanomolar affinity have been obtained for tetrodotoxin [55,56], while the scFvs obtained against tetrodotoxin have affinities in the micromolar range [57,58].

In terms of safety, antibodies and aptamers face similar issues, but aptamers have the advantage in certain areas. As with all affinity reagents, off-target binding is a concern for both. For antibodies, this can be mitigated by using monoclonal antibodies with defined binding sites and testing them in vitro and in vivo for reactivity to other targets [59]. While this is true for aptamers as well, negative selection can also be used during SELEX to remove cross-reactive and non-specific aptamers from the oligo pool before they are sequenced and individually tested [21]. On-target effects can also be a concern for both, as binding to normal human proteins, such as TNF-α [60], can inhibit both pathological and beneficial functions. The main advantage of aptamers is that they are typically non-immunogenic and non-toxic [61,62]. This can help prevent the immune-mediated reactions seen in antibody therapies [63] and means that they are typically very well tolerated with minimal side-effects [62].

However, there are areas in which antibodies outperform aptamers. While they are more stable outside of the body, aptamers tend to have much shorter half-lives in vivo due to nuclease digestion [64] and renal filtration [65]. Without modification, half-lives can be as short as minutes or hours and even with modifications to prevent renal filtration and nuclease digestion, half-lives are typically 1–2 days and occasionally a couple weeks [62]. In contrast, humanized antibodies with modified Fc (fragment crystallizable) regions can have half-lives upwards of two months [66]. While this may not be as impactful in treatments for acute diseases, this could have significant implications for long-term treatments and prophylaxis. Additionally, the constant regions of human or humanized antibodies can interact with other immune components, such as phagocytic cells and complement components, which can enhance the therapeutic response [67].

## 4. Clinical Trials and Approved Drugs

### 4.1. Therapies for Ocular Diseases

Aptamer therapies have had significant success in the treatment of ocular diseases, especially macular degeneration [68]. Age-related macular degeneration (AMD) is an ocular disease that causes progressive vision loss in the elderly [69]. It can be categorized into wet AMD (also known as exudative or neovascular AMD) or dry AMD (also known as atrophic AMD) based on pathology [69]. Some of the principal manifestations of wet AMD include the detachment of the retinal pigment epithelium, macular hemorrhaging and edema, and the abnormal growth of choroidal vasculature in the eye, called choroidal neovascularization (CNV) [69]. Dry AMD is primarily characterized by atrophy in the eye, called geographic atrophy (GA) [69]. Dry AMD is more common, representing 90% of AMD cases, but wet AMD is typically more severe [69]. The aptamers tested for the treatment of AMD in clinical trials have one of three targets: vascular endothelial growth factor (VEGF), platelet-derived growth factor (PDGF), and the complement component C5.

There are various members of the VEGF gene family and numerous isoforms of those members, many of which have important physiological functions, including acting as survival factors for neurons [70]. Therefore, aptamer development was focused on VEGF-165, the isoform that is thought to contribute the most to the neovascularization-related pathology of wet AMD, to limit adverse effects from treatment [68]. This resulted in the development of Macugen, a pegylated RNA aptamer that was approved by the FDA due to two double-blind, placebo-controlled studies that demonstrated moderate efficacy in preventing vision loss when injected intravenously every 6 weeks for 48 weeks [71]. A follow-up study demonstrated that patients receiving Macugen had a 45% relative benefit over placebo controls in mean changes in vision after 102 weeks [72]. Furthermore, the drug was well tolerated in humans and animals, and no antibodies were detected against Macugen in the clinical and pre-clinical trials [71,72,73]. Other applications of Macugen have also been explored, including diabetic macular edema [74].

Similar to VEGF, PDGF helps promote neovascularization in wet AMD, and has therefore been a drug target for treatment [75]. This resulted in the development of Fovista, a pegylated DNA aptamer targeting PDGF, which was tested in combination with the anti-VEGF therapy ranibizumab but showed negative patient outcomes compared to ranibizumab alone [75].

The other FDA-approved drug, Izervay, previously known as Zimura, is a pegylated RNA aptamer targeting the complement component C5 for the treatment of GA in dry AMD [76]. It is thought to work by preventing overactive immune response and inflammation due to C5 activation, which can promote atrophy in dry AMD [77]. In the double-blind, placebo-controlled phase II/III study, patients who were intravitreally treated once monthly for 12 months showed a 35% reduction in mean lesion growth compared to control [76]. The most reported adverse effects were conjunctival hemorrhage, increased intraocular pressure, blurred vision, and CNV, with adverse effects reported in 52.2% of 2 mg recipients, 68.7% of 4 mg recipients, and 34.5% of placebo recipients [76]. Additionally, Izervay is currently being evaluated as a treatment for Stargardt disease [78].

### 4.2. Autoimmune, Cancer, and Infectious Disease Therapies

Another popular application for aptamers is immunomodulation. Three aptamers in clinical trials target chemokines, thereby affecting the migration and recruitment of immune cells. The mixed L-RNA/L-DNA aptamer AON-D21 targets C5a and C5adesArg, cleavage products of complement component C5 that are important for inflammatory responses, as they recruit pro-inflammatory immune cells, activate phagocytic cells, and trigger granule release [79,80]. This inflammation can lead to immunopathology in some cases, so AON-D21 is being explored as an infectious disease therapeutic and is undergoing clinical trials for the treatment of community-acquired pneumonia [81]. AON-D21 is also in clinical trials for the autoimmune disease paroxysmal nocturnal hemoglobinuria (PNH) since C5a and C5adesArg are also involved in the migration of hematopoietic stem/progenitor cells (HSPCs) from the bone marrow to the peripheral blood, which is important for the pathogenesis of PNH [82,83]. Conversely, CXCL12 (SDF-1) is a chemokine that helps retain immune cells in the bone marrow and is the target of NOX-A12, an L-RNA aptamer [84,85]. By inhibiting CXCL12, NOX-A12 promotes the mobilization of immune cells from the bone marrow, which may be beneficial in treating myelomas and leukemias by making them more accessible to other therapies and is currently in clinical trials for relapsed/refractory multiple myeloma [85,86,87]. The administration of NOX-A12 during HSPC transplant to allow for harvesting the cells from peripheral blood instead of bone marrow is also being explored in clinical trials [88]. Another chemokine-targeting aptamer is NOX-E36, which inhibits CCL2 (MCP-1) [89]. CCL2 is important in macrophage and monocyte migration, which can lead to inflammatory responses that are involved in diabetes, lupus, and other kinds of chronic inflammation, so NOX-E36 is therefore also in clinical trials [89,90,91]. Interestingly, the combination of NOX-E36 and NOX-A12 showed promising results in the treatment of murine proliferative lupus nephritis [92].

Aside from their use as chemokine antagonists, aptamers can have other applications in the treatment of infections and cancer. ApTOLL is a DNA aptamer that is undergoing clinical trials for the treatment of COVID-19 [93]. It inhibits TLR4, an important pathogen recognition receptor involved in inflammatory responses that can be damaging in certain infectious diseases [94,95]. AS1411 is an aptamer undergoing clinical trials for the treatment of relapsed/refractory multiple acute myeloid leukemia and renal cell carcinoma [96]. It is unique in this list in that it does not bind to a component of the immune system. It instead binds nucleolin, which is a protein that is overexpressed in the cytoplasm and on the cell surface of certain types of tumor cells [97]. The cytoplasmic nucleolin can stabilize bcl-2 mRNA, leading to increased Bcl-2 expression in cancer cells, which helps prevent apoptosis [97,98]. The model for the mechanism of action of AS1411 is therefore that increased expression of nucleolin on the surface of tumors enhances the uptake of AS1411 into the cytoplasm, where it can then bind cytoplasmic nucleolin and prevent it from stabilizing bcl-2 mRNA, leading to increased apoptotic cell death [96].

### 4.3. Therapies for Blood Disorders

Another common clinical application of aptamers is in the treatment of blood disorders and clotting. In addition to its role in infectious disease treatment, ApTOLL is also undergoing clinical trials to prevent excess inflammation during the acute phase of ischemic strokes, where it demonstrated safety and efficacy in Phase I/II [99]. Similarly, the aptamer-based system REG1 is being studied as an anticoagulation agent in acute coronary syndrome [100]. It includes the modified RNA aptamer pegnivacogin, which inhibits Factor IXa, a component of the clotting cascade [100]. Interestingly, the REG1 system also includes anivamersen, an RNA oligonucleotide inhibitor of pegnivacogin, which acts through base pairing and is important for preventing excess bleeding that can result from anticoagulation therapy [100].

On the opposite side of the spectrum, aptamers have also been developed to treat bleeding disorders like von Willebrand Disease (VWD) and hemophilia. One approach for the treatment of these disorders is to increase the half-life of the von Willebrand factor (VWF). VWF is a chaperone for coagulation factor VIII, but it can be cleared from the bloodstream via interaction between its A1 domain and LRP-1 receptor on macrophages, reducing the serum half-life of both VWF and factor VIII [101]. Therefore, the inhibition of the A1 domain of VWF can lead to increased levels of factor VIII, which could be beneficial for hemophilia A, where there is a factor VIII deficiency, and type 2B VWD, where the A1 domain of VWF has increased activity [102,103,104]. To this end, the aptamer ARC1779 was developed against the A1 domain of VWF. While it reached Phase I/II clinical trials, it suffered from a short serum-half life that reduced its effectiveness [105]. Therefore, the aptamer BT200 was developed, which has a longer serum-half life, good bioavailability, and is currently in clinical trials with promising results thus far [104,106]. Another approach is the inhibition of the tissue factor pathway inhibitor (TFPI). TFPI inhibits coagulation factors VIIa and Xa, so the downregulation of TFPI can promote clotting and could be beneficial in hemophilia patients [107]. For this approach, the TFPI-binding aptamer ARC19499 was developed [108]. ARC19499 has shown promise in a monkey model of hemophilia and reached clinical trials, though the first-in-human trial was unfortunately terminated (NCT01191372) [108].

Another aptamer developed for the treatment of blood disorders is NOX-H94, which binds to and inhibits hepcidin [109]. Hepcidin is an antimicrobial peptide and regulator of iron homeostasis central to the pathogenesis of anemia of chronic inflammation (ACI) [110,111]. By inhibiting hepcidin, NOX-H94 treatment has shown promise in an acute cynomolgus monkey model of IL-6-induced hypoferremia [109] and a Phase I/II clinical trial in experimental human endotoxemia [112]. It has also completed a Phase I/II trial in dialysis patients with erythropoiesis stimulating agents (ESA)-hyporesponsive anemia, though the results are not yet available (NCT02079896).

## 5. Future Directions

### 5.1. Therapies for Small Molecule Toxins

There are a variety of small molecule toxins that cause numerous human intoxications and deaths due to environmental exposures. Of them, mycotoxins and marine toxins are some of the most impactful [113]. Mycotoxins are produced by fungi that can contaminate food, especially when improperly stored, and can cause human diseases and deaths [114]. According to the FAO, it is estimated that over 25% of the agricultural products of the world are contaminated with one or more of these toxins, and it is especially problematic in developing countries, such as Africa [114]. The ingestion of these toxins can cause a variety of diseases, including nephropathy, kidney damage, immunosuppression, and cancer [114]. Phycoxins are marine toxins that are primarily produced by algae and accumulate in a variety of aquatic invertebrates, including shellfish, which is why they are commonly known as shellfish toxins [115]. Depending on the specific toxin ingested, these can cause paralytic shellfish poisoning (PSP), amnesic shellfish poisoning (ASP), or diarrhetic shellfish poisoning (DSP) [115]. These toxins are responsible for over 60,000 human intoxications a year and cause hundreds of millions of dollars in economic damage due to their negative impact on the shellfish industry, recreational activities, and tourism [115]. Additionally, as some of these toxins are rapidly lethal in low-doses and there are limited treatment options available, they are also considered potential bioterror weapons [116,117].

Antibodies are one of the classical treatments for toxins, but they are especially difficult to produce for small molecule toxins. One reason is that many of these toxins are highly lethal in low doses, making the acquisition of neutralizing antibodies from animal immunizations difficult. Furthermore, small molecules are poorly immunogenic and often require carriers for immunization, which can lead to a variety of challenges, including the shielding of the antigen by the carrier [46]. Despite these challenges, monoclonal antibodies and antibody fragments targeting small molecule toxins, such as tetrodotoxin (TTX), have been produced [57,58,118]. However, the only monoclonal antibody produced for TTX showed little therapeutic efficacy in humans [119]. The three single-chain antibody fragments produced for TTX (scFv-T53, s35-HuScFv, and s16-HuScFv) also have had challenges for effective use. scFv-T53 has relatively low affinity—1.1 × 10^6^ M^−1^—and has not been tested in humans or animals for efficacy [57]. The other two require a 4:1 molar ratio or above of antibody to TTX for neutralization in vitro and even higher molar ratios were required in mice, as even a ratio of 7:1 only delayed the death when the animals were dosed with 1.5 MU (0.3 ug TTX and 180 ug of scfv) [58].

Aptamers show promise as a potential alternative to antibodies in this application. For one, they are small and quite flexible, so they are capable of binding small molecules with high affinity [120]. In fact, multiple aptamers have been developed with nanomolar affinity to TTX [55], with a hundreds of times greater affinity than that of the scFv-T53 antibody fragment. The ease of production and ability to produce many aptamers in parallel, as described earlier, would also aid in the production of cocktails active against multiple toxins. This is especially important since many toxins are fast-acting, so treatment without full diagnostics would be beneficial. While producing these cocktails against many different toxins would be challenging, this process would be expedited by testing and modifying pre-existing aptamers, especially once aptamers for related toxins are developed. This could be undertaken with structural predictions, as has been carried out previously for identifying aptamers for TTX [55], or through the random mutagenesis of aptamers for related toxins. Finally, the long-shelf life and thermal stability of aptamers is a significant advantage when stockpiling therapeutics for biodefense.

A major limitation to this application is that while many aptamers have been developed to bind small molecule toxins, the neutralization ability of toxin-targeting aptamers is rarely tested, as their most common application to date is detection [121,122]. However, aptamers have been developed to effectively neutralize αC-conotoxin PrXA, a small peptide toxin and potential bioweapon, as well as the endotoxin LPS, indicating that they have to potential to neutralize other toxins as well [123,124]. With more research into this area, aptamers could become an important therapeutic for a variety of toxins for which we currently have very few treatment options.

### 5.2. Combinatorial Approaches

Aside from the inhibition-based approaches described above, aptamers are also effective tools for the delivery of other drugs. As mentioned previously, they are non-immunogenic and small, which allows an easier delivery across physiological barriers. In fact, there are in vivo SELEX methods for selecting aptamers that can penetrate the blood–brain barrier and accumulate in the brain [125]. Additionally, approaches such as cell-internalization SELEX allow for the selection of aptamers that selectively enter certain cell types, which can be used to deliver drugs to those cells [126]. One promising disease target for this is cancer, as cancer cells often have unique expression profiles of cell surface proteins and can express novel proteins, called neoantigens [127]. For example, in addition to its direct anticancer effects, AS1411 is also being incorporated into nanoparticles for drug delivery to cancer cells [128]. Interestingly, the incorporation of AS1411-containing sequences into gold nanoparticles also significantly enhanced the accumulation of the aptamer in the cells and its anticancer activity [129].

Aptamers are especially apt for the delivery of other nucleic acid-based therapeutics, as they can be incorporated into the same nucleic acid molecule and synthesized in vitro. A popular approach is the delivery of small interfering RNAs (siRNAs) to certain cell types, which can then silence the expression of target genes within those cells [126]. This has many applications, including the development of antiviral agents. For example, CD4 aptamer-siRNA chimeras are capable of suppressing HIV replication in cell culture and tissue explants by reducing the expression of CCR5 or HIV genes in macrophage CD4+ T cells, the host cells for HIV [130]. Methods for generating multivalent aptamers have also been developed to increase the affinity of the aptamer constructs of the target up to 200-fold [131]. Aptamers could theoretically also be used to deliver ribozymes and DNAzymes, single stranded nucleic acid-based enzymes that have also been explored as therapeutics, including in antiviral and anticancer applications [132,133]. A potential drawback of this approach is that both aptamers and ribozymes/DNAzymes have extensive secondary structures that are essential for their function, so chimeras would have to be carefully designed to prevent the disruption of those structures. However, a search of the literature revealed no articles in which aptamer–ribozyme/DNAzymes have been constructed, so it is unclear to what extent this would be an issue. Despite the potential challenges, the generation of these chimeras is a promising avenue of research that may lead to more effective therapeutic compounds than any of the agents alone.

## 6. Conclusions

As highlighted throughout this review, aptamers have significant promise as therapeutic agents. Their in vitro selection process, ease of production, low immunogenicity, and various other aspects provide unique advantages not offered by antibodies. While there are also important drawbacks, especially their low in vivo stability, many different approaches have been developed to combat these challenges, which have resulted in a number of successful pre-clinical studies, clinical trials, and two FDA-approved drugs in a wide variety of applications, including the treatment of ocular diseases, cancers, infectious diseases, autoimmune diseases, bleeding disorders, clotting, and anemia. However, there is still significant research that can be carried out to develop novel aptamer-based therapeutics that capitalize on their unique advantages, including antitoxins and combinatorial therapies with nanoparticles and other nucleic acid therapeutics. Research in these and other areas has the potential to significantly advance the field, provide more effective treatments, and expand the range of clinical applications beyond what is currently available.

## Figures and Tables

**Figure 1 ijms-25-06742-f001:**
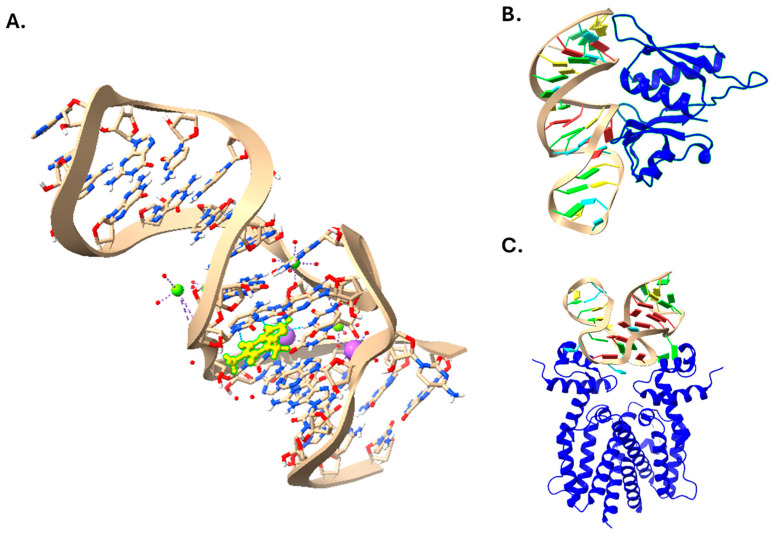
Structures of aptamer–ligand complexes. (**A**) Aptamer in complex with theophylline (yellow) and magnesium (green) and sodium (purple) ions. Light blue dashed lines represent hydrogen bonding, darker dashed lines indicate ionic bonding. PDB: 8D28. Aptamers are also shown in complex with *Bacillus anthracis* ribosomal protein S8 (**B**; PDB: 4PDB) and TetR (**C**; PDB: 6SY6), both shown in blue.

**Figure 2 ijms-25-06742-f002:**
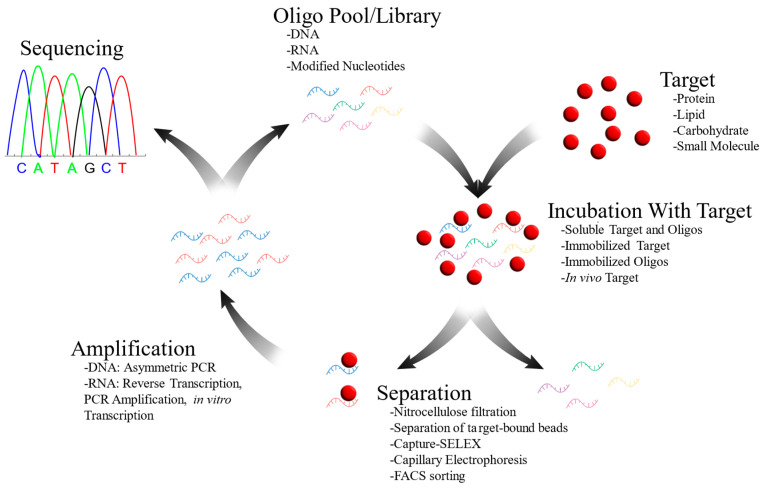
Graphical illustration of SELEX procedure with example methodologies and variations are included at each step. Red and blue oligos represent target-binding aptamers, while all other colors represent non-binding oligos.

## Data Availability

No new data were created or analyzed in this study. Data sharing is not applicable to this article.

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
