# Peer review of "Therapeutic Applications of Aptamers"

_ijms, 2024, doi:10.3390/ijms25126742_

Round 1
Reviewer 1 Report
Comments and Suggestions for Authors
The authors summarize research on the medical applications of aptamers. Though the section on methodology for aptamer selection is more verbose in view of the title and the aim of this manuscript, overall, I can appreciate that it is well-written. Reference papers are appropriately selected from the past to the present. I agree with the publication of this review article in the International Journal of Molecular Sciences after a very minor revision that responds to my following comments.
With respect to the statement "The specificity of aptamers does not correlate well with their affinity, meaning higher affinity aptamers may not necessarily be more specific for the target." in line 40, A.D. Ellington previously described "Tighter binding means higher specificity. The tighter binder having more contacts that are formed with a ligand, the more potential sites of discrimination there will be." I agree with him because the difference in enthalpy change during complex formation may bring specificity. What do you think about that?
Incomplete references 3, 5, 11, 18, 22, 39, 40, 48, 60, 64, 67, 104, 118, and 119 must be fixed.
-80 C in L.249 should be -80°C. Also, 1.1 x 106 M-1 in L.435 has to be correctly styled.
Author Response
Thank you for your review.
Regarding specificity and affinity, I agree that they are usually well-correlated and could be correlated within derivatives of a single aptamer. However, as Carothers and colleagues demonstrated (citation 9), this does not appear to hold up experimentally with independently selected aptamers. The authors of that paper attributed this to the aptamers having unique interactions with their target that result in different specificity patterns, though they were also not expecting these results.
I have corrected the citations and the errors on lines 249 and 435.
Reviewer 2 Report
Comments and Suggestions for Authors
In the Review Article entitled "Therapeutic Applications of Aptamers” by George Santarpia and Eric Carnes, the authors reviewed the recent literature on the synthetic natural and modified nucleic acids analogs in therapeutic applications. The topic is very appealing, particularly considering the recent interest in RNA drug design. The review is well organized and easy to follow. In my opinion, once it has been subjected to minor revisions, this work will be suitable for publication in “International Journal of Molecular Sciences”.
Minor comments:
1) Line 102: “confirmation” should be “conformation.”
2) In the paragraph: 2.2 Binding analysis and optimization, among the post-selex chemical modifications should also be reported the introduction of inversion of polarity site, since the 3’-capping with inverted thymidine is commonly used to increase the aptamer resistance to 3’-exonucleases and to prolong their half-life time in serum (see Macugen).
3) The reference list is quite up to date, however many references are incomplete, so I recommend careful review.
Author Response
Thank you for your review.
We have corrected the error on line 102 and the citations. We also added a mention of capping the 3' and 5' ends for nuclease resistance. However, we did not specify capping with inverted thymidine as there are other variations, including biotin-streptavidin caps. The sentence currently reads: "Some modifications – such as replacement of the phosphodiester backbone, additions to the 2’ or 4’ carbon of the ribose, and capping of the 3’ or 5’ end – decrease recognition by nucleases [29][30]." Is that acceptable?